# Opioid-Induced Respiratory Depression in Pediatric Palliative Care Patients with Severe Neurological Impairment—A Scoping Literature Review and Case Reports

**DOI:** 10.3390/children7120312

**Published:** 2020-12-21

**Authors:** Maximilian David Mauritz, Carola Hasan, Larissa Alice Dreier, Pia Schmidt, Boris Zernikow

**Affiliations:** 1Paediatric Palliative Care Centre, Children’s and Adolescents’ Hospital, 45711 Datteln, Germany; c.hasan@kinderklinik-datteln.de (C.H.); p.schmidt@kinderpalliativzentrum.de (P.S.); b.zernikow@kinderklinik-datteln.de (B.Z.); 2Department of Children’s Pain Therapy and Paediatric Palliative Care, Faculty of Health, School of Medicine, Witten/Herdecke University, 58448 Witten, Germany; 3PedScience Research Institute, 45711 Datteln, Germany; l.dreier@pedscience.de

**Keywords:** opioid-induced critical respiratory events, Opioid-Induced Respiratory Depression, OIRD, Severe Neurological Impairment, children, pain, palliative care

## Abstract

Pediatric Palliative Care (PPC) addresses children, adolescents, and young adults with a broad spectrum of underlying diseases. A substantial proportion of these patients have irreversible conditions accompanied by Severe Neurological Impairment (SNI). For the treatment of pain and dyspnea, strong opioids are widely used in PPC. Nonetheless, there is considerable uncertainty regarding the opioid-related side effects in pediatric patients with SNI, particularly concerning Opioid-Induced Respiratory Depression (OIRD). Research on pain and OIRD in pediatric patients with SNI is limited. Using scoping review methodology, we performed a systematic literature search for OIRD in pediatric patients with SNI. Out of *n* = 521 identified articles, *n* = 6 studies were included in the review. Most studies examined the effects of short-term intravenous opioid therapy. The incidence of OIRD varied between 0.13% and 4.6%; besides SNI, comorbidities, and polypharmacy were the most relevant risk factors. Additionally, three clinical cases of OIRD in PPC patients receiving oral or transdermal opioids are presented and discussed. The case reports indicate that the risk factors identified in the scoping review also apply to adolescents and young adults with SNI receiving low-dose oral or transdermal opioid therapy. However, the risk of OIRD should never be a barrier to adequate symptom relief. We recommend careful consideration and systematic observation of opioid therapy in this population of patients.

## 1. Introduction

Pediatric Palliative Care (PPC) accompanies children, adolescents, and young adults with a broad spectrum of life-threatening or life-limiting conditions (LTC/LLC) for many years or even decades [1,2]. In contrast to adult palliative care patients, a substantial proportion of these patients have irreversible but non-progressive conditions that originated in the perinatal or postnatal period [3]. Progressive conditions in this population are neurodegenerative or metabolic conditions without curative treatment options [4,5,6]. Typical examples are hypoxic-ischemic encephalopathy, severe cerebral palsy, congenital brain malformations, and progressive syndromes, such as those caused by chromosomal abnormalities or metabolic disorders like mucopolysaccharidosis. Additionally, many PPC patients experience Severe Neurological Impairment (SNI), defined by a combination of motor impairment, medical complexity, and the need for assistance with activities of daily living, resulting from the above-mentioned disorders of the central nervous system [7].

Pain is a frequent symptom in PPC patients with SNI and seems to be most prevalent in those with the most significant impairments [4,8]. In PPC patients with non-cancer LTC/LLC, common causes of pain are musculoskeletal pain, muscle spasticity, and hip dislocation or scoliosis caused by contractures. If causal therapy for those conditions fails or is not indicated, analgesic medication is initiated for adequate and permanent symptom control in PPC patients [9]. Strong opioids are frequently used in PPC for treating chronic, acute, nociceptive, or neuropathic pain, but also other distressing symptoms such as dyspnea [9,10,11]. Besides immediate acting intravenous, subcutaneous, and mucosal preparations, strong opioids are also available as oral sustained-release and transdermal applications. Adverse effects of strong opioids include constipation, nausea/vomiting, pruritis, fatigue, confusion/hallucination, myoclonus, urinary retention, and rarely—but most severely—Opioid-Induced Respiratory Depression (OIRD) [9]. Nevertheless, their safety profile makes them preferable to non-steroidal anti-inflammatory drugs (which may induce renal, gastric, and hepatic side effects [12,13]) and weak opioids (which can cause nausea or seizures [14], especially with long-term use [9,11]).

Research on pain and OIRD in pediatric patients with SNI is scarce. While a considerable proportion of these patients experience chronic pain as well as postoperative pain due to frequent surgical procedures, they are often excluded from studies on pediatric pain management [15,16,17]. Among clinicians, uncertainties exist regarding pain management and the risk of adverse effects in this vulnerable patient group [18,19]. The few data available on the assessment and management of pain in patients with SNI and their risk of OIRD particularly focus on short-term postoperative intravenous opioid analgesia via Patient-Controlled Analgesia (PCA) or PCA by Proxy (PCAP) [20]. Despite representing the preferred administration route for long-term symptom control in PPC, case reports of OIRD due to oral or transdermal opioids in this patient population are limited [17].

The pathophysiological process of Opioid-Induced Respiratory Depression is well understood. The respiratory drive is generated in the brainstem. It is influenced by inputs from the cortex, as well as by central and peripheral chemoreceptors. The clinically relevant analgesic effect of opioids is mediated mainly by the mu-receptor, with potential effects on delta-opioid and kappa-opioid receptors [21]. Mu-receptor activation also seems to be solely responsible for opioid respiratory effects [20,22]. Binding to the mu-receptors activates inhibitory intracellular pathways, leading to an attenuation of the brainstem’s ventilatory responses to hypoxia and hypercapnia [22,23]. Even small opioid doses will affect a patient’s breathing pattern, and higher doses may lead to a decline in the patient’s respiratory rate. Although changes to tidal volume and minute ventilation may maintain an adequate blood oxygen saturation, increased dosage deteriorates these compensatory mechanisms that may potentially result in respiratory failure [24].

This study aims to present a comprehensive summary of currently available research on OIRD in pediatric patients with SNI by means of a scoping review. We subsequently present three clinical cases of severe respiratory events in PPC patients with SNI who were treated with oral or transdermal opioids for pain management in a specialized PPC unit in Datteln, Germany. To our knowledge, these are the first reported cases of OIRD after oral and transdermal administration in pediatric patients with SNI.

## 2. Materials and Methods

This study complies with the “Preferred Reporting Items for Systematic reviews and Meta-Analysis extension for Scoping Reviews (PRISMA-ScR)” guidelines [25]. For the scoping review, a literature search was conducted for articles reporting on OIRD in pediatric patients with SNI. Since no common definition for OIRD exists, for the study purpose, OIRD was defined as any opioid-related significant decrease in the respiratory rate or oxygen desaturation that led to apnea or the need for medical intervention. Regarding SNI, we followed the latest delphi consensus-based definition [7]. On the 14th of October 2019, a Medline literature search identified English and German language studies. The search strategy is presented in Table 1. The search terms included “opioids” in general and the specific opioids commonly used in PPC in Germany. In addition to the electronic database search, articles were further identified by hand search.

Articles eligible for inclusion were required to report on OIRD in children and adolescents with SNI. Exclusion criteria were defined as (1) age less than six months, (2) age over 21 years, (3) children with no SNI, and (4) no OIRD. Respiratory support or tracheostomy did not per se lead to patient exclusion. Patients with respiratory support or tracheostomy were not excluded. After excluding duplicate articles, the remaining articles’ titles and abstracts were blindly screened for inclusion by two authors (MM and PS). MM and PS also independently screened the full texts of all eligible articles. Any discrepancies in the assessment of inclusion and exclusion criteria were discussed until a consensus was achieved.

## 3. Results

### 3.1. Selection of Sources of Evidence

The electronic database search yielded 440 articles. An additional 104 articles were identified through hand searching, and 521 articles remained after duplicates were removed. Based on the initial abstract screening, nine articles remained for further full-text assessment. Of these articles, a total of six met eligibility criteria and were included in the scoping review (see Figure 1).

Table 2 shows detailed information on the six included articles. Four of these articles reported on retrospective data [26,27,28,29], while two articles reported on prospective data [30,31]. Four studies were conducted in the United States of America [26,28,29,31], one in the United Kingdom [27], and one in the United Kingdom and the Republic of Ireland [30]. Two studies [27,28] specifically investigated SNI as a potential risk factor for OIRD in pediatric patients; four studies focused on risk factors for OIRD in general.

The reviewed literature reported adverse respiratory events with different severity levels. Opioid-Induced Critical Respiratory Events or OIRD were defined as an adverse event leading to the administration of naloxone [29], airway management/escalation of care [26], or death/harm [27,30].

Five of the six studies examined the incidence and risk factors for OIRD during short-term intravenous PCA [26,29], Patient/Nurse-Controlled Analgesia (PNCA) [28,30], Nurse-Controlled Analgesia (NCA) [27] or PCAP [26]. One study also reported OIRD after short-term oral opioid or combined intravenous/oral administration [29].

### 3.2. Incidence of Opioid-Induced Respiratory Depression Events in Children

All included studies reported on the incidence of OIRD in children predominately receiving short-term postoperative opioid medication. The reported incidence of OIRD varied between 0.13% and 4.6% due to the different contexts of application. Only Jay et al. [27] named a concrete incidence of OIRD in children with SNI. In their study population of children receiving NCA, the overall incidence of respiratory depression was 0.68%, and the cumulative incidence of OIRD in the SNI group was 1.09% vs. 0.59% in the control group.

Respiratory depression and somnolence, leading to the administration of naloxone occurred in 2.8% of all patients reported in the article by Czarnecki et al. [28]. Voepel-Lewis et al. [26] reported OIRD requiring naloxone administration, airway management, or escalation of care in 4.6% of patients receiving postoperative PCAP. In the article by Monitto et al. [31], 1.7% of patients needed naloxone to treat PNCA-related apnea or desaturation. In Chidambaran et al. [29], 0.06% of all patients undergoing opioid treatment required naloxone. In their national audit, Morton et al. [30] reported an overall incidence of serious clinical events of 0.43% and a respiratory depression incidence of 0.13%.

### 3.3. Risk factors of OIRD

#### 3.3.1. Severe Neurological Impairment

Most authors identified SNI as an independent risk factor for OIRD [26,27,28,29,30]. Chidambaran et al. [29] identified developmental delay and neurological impairment as significant risk factors for early respiratory depression caused by opioids. The unadjusted odds ratio for the necessity of naloxone administration was 3.24 (CI 1.36–7.47) for the presence of an underlying syndrome, 4.99 (CI 2.17–11.15) for developmental delay, and 3.87 (CI 1.83–8.07) for neurological dysfunction. Morton et al. [30] presented 14 individual patient reports concerning respiratory depression; eight patients received naloxone, all of whom were either very young or had significant neurodevelopmental, respiratory, or cardiac comorbidities. Focusing on children with neurodevelopmental disorders, Jay et al. [27] described a higher risk of respiratory depression with increasing opioid doses in their SNI group vs. the control group. Additionally, there was a significant difference between the cumulative incidence of OIRD in the control group and the SNI group. In this latter group, children with cerebral palsy, Down’s syndrome, and encephalopathy showed an increased risk of developing both respiratory depression and other serious adverse events.

Monitto et al. [31] found no significant risk factor associated with naloxone administration in a sample of nonsurgical and postoperative children below six years of age. The study identified individual clinical characteristics as predisposing factors for patients’ excessive sedation or respiratory compromise, though solely developmental delay, congenital anomalies, and additional sedatives were described as specific risk factors.

#### 3.3.2. Polypharmacy, Comorbidities, and Additional Risk Factors

Five articles reviewed risk factors for respiratory depression after opioid administration [27,28,29,30,31]. Younger children (<one year of age) were at greater odds of respiratory depression than older children [27,29,30]. Four articles [27,28,29,31] determined that polypharmacy or sedative comedication was a risk factor for OIRD; the additive effects of sedatives (e.g., benzodiazepines) when co-administered with opioids may cause a central nervous system depression. Furthermore, in their study of postoperative pediatric patients with SNI, Czarnecki et al. [28] reported a potential association between adjuvant sedating medications (diazepam, droperidol, chloral hydrate, and diphenhydramine) and respiratory depression. Medication errors led to adverse effects in three studies [29,30,31].

Morton et al. [30] similarly described the concurrent administration of sedatives and opioids, as well as additional respiratory and cardiac comorbidities as risk factors for OIRD. Patients with SNI frequently have impaired respiratory drive, obstructive sleep apnea, cardiorespiratory deficits, neuromuscular and postural abnormalities, and frequent gastroesophageal reflux as co-factors [27] of OIRD. Additional risk factors include prematurity, obesity, hepatic dysfunction, and major airway surgery [29].

#### 3.3.3. Preventive Strategies

The authors recommend some potential preventive strategies for avoiding OIRD in children. Four studies advised against co-administration of opioids and CNS depressants or, if this should not be possible, recommended a significant opioid dose reduction [28,29,30,31]. Adequate monitoring, including pulse oximetry and close monitoring of sedation depth, was emphasized in three studies [26,29,31]. Two authors recommended special consideration of additional risks triggered by comorbidities of the underlying diseases [29,30].

### 3.4. Case Reports

Three cases of OIRD in children with SNI treated in a specialized PPC unit are described hereinafter. The PPC unit is a self-reliant unit in a tertiary care children’s hospital, comprising eight patient beds in single rooms and offering intensive inpatient PPC. Annually, approximately *n* = 200 children with LTC/LLC and a mean age of 9.5 years are admitted to the unit. This first PPC unit in Europe opened in 2010. It is located in one of the three largest pediatric hospitals in Germany. The idea to conceptualize a specialized pediatric palliative care unit derived from ten years’ experience in specialized pediatric palliative home care in the most crowded are of Germany. The PPC home care team was often confronted with therapy refractory symptoms that required a more intensive bio-psycho-social assessment and treatment of a multidisciplinary team. The PPC unit consists of eight single bedrooms with garden access. Parents can accompany their children. They can sleep in the same bedroom or in a separate family apartment on the second floor. The professional team includes pediatricians of diverse medical backgrounds (oncology, neonatology, intensive care medicine), nurses with various specialist trainings (wound care, respiratory medicine, etc.), psychologists, and other therapists (e.g., music therapists). Patients mainly experience irreversible but non-progressive conditions originating in the perinatal or postnatal period, or progressive neurodegenerative or metabolic conditions. Pain is a common symptom at the time of admission, present in 47% of all patients [6,32]. In other hospitals, these children are possibly treated in units such as the general pediatrics unit, the pediatric neurology unit, or the intermediate care unit. In some countries, dose titration with strong opioids in children with SNI is performed outside the hospital, for example, in children’s hospices.

### 3.5. General Information

During opioid titration, children were monitored by pulse oximetry and regular measurement of their respiratory rate. The patients’ characteristics are specified in Table 3.

#### 3.5.1. Case 1

The first case report refers to an 18-year-old young adult with hypoxic-ischemic encephalopathy and spastic cerebral palsy (GMFCS level V) following preterm birth and who received cardiopulmonary resuscitation due to severe sepsis at the age of 16 years. Owing to the presence of severe musculoskeletal pain caused by spasticity and hip luxation, opioid therapy with an ultra-low dose of 2 mg oral morphine four times a day was started via the patient’s gastric tube. On day two of treatment, he showed an oxygen desaturation (lowest peripheral oxygen saturation was 82%) and a reduced respiratory rate of 7/min, requiring repeated stimulation. The daily oral Morphine Equivalent Dose (MED) was 0.241 mg/kg/d, equivalent to 16.87 mg/d in a 70 kg patient. At this time, the patient was severely underweight (BMI 16.8 kg/m², 2nd percentile) and was concomitantly treated with 13 different substances, including levetiracetam, valproate, dronabinol, and chloral hydrate. As a result, the morphine therapy was discontinued. Two days post-incident, the patient developed pneumonia, necessitating antibiotic treatment. A second opioid-based pain therapy was attempted during the remainder of the inpatient course. We decided to attempt opioid treatment with levomethadone. Oral levomethadone was titrated up to 0.5 mg twice daily under close monitoring of pulse oximetry and respiratory rate until sufficient pain relief was achieved. The patient did not show any signs of respiratory depression under oral levomethadone therapy.

#### 3.5.2. Case 2

The second patient is a 15-year-old adolescent with early infantile epileptic encephalopathy (Ohtahara syndrome) and spastic cerebral palsy (GMFCS level V). He was admitted to the PPC unit to evaluate and manage his severe neuroirritability associated with pain-like behavior and seizures. Because of a possible central, neuropathic pain component of the therapy-refractory agitation states, we started oral levomethadone therapy of 2.5 mg three times daily via the patient’s gastric tube. The treatment was paused after one day of treatment due to increased myoclonus, seizures, and oxygen desaturations requiring oxygen supply. At the time of the incident, the patient received a comedication of 10 substances, including lamotrigine, domperidone, and risperidone. A second attempt with 1 mg of levomethadone orally three times daily was conducted. On day 8 of therapy, the patient showed increased apnea phases and oxygen desaturations, requiring repeated stimulation and oxygen supply (oral MED: 1.02 mg/kg/d [33], equivalent to 71.4 mg/d in a 70 kg patient). Further into the inpatient course, a decrease in the frequency and extent of painful agitation was achieved using an even slower dose-titration of levomethadone up to 3.25 mg orally three times a day.

#### 3.5.3. Case 3

The third patient is a 20-year-old young adult with lissencephaly and spastic cerebral palsy (GMFCS level V) having a cognitive developmental age of less than 12 months. The patient was severely underweight (BMI 13.8 kg/m², <1st percentile), was concomitantly treated with 12 different substances, including oxcarbazepine and valproate, and had severe obstructive sleep apnea. He was experiencing severe musculoskeletal pain secondary to spasticity as well as hip dysplasia refractory to oral therapy with the weak opioid tilidine, administered in a fixed combination with naloxone (this fixed combination is supposed to prevent the drug’s intravenous abuse). In the absence of a gastric tube, a therapeutic attempt was initiated with transdermal buprenorphine (5 µg/h patch) as a long-acting alternative to intensified oral opioid therapy. On the night of the first day of therapy, the patient showed oxygen desaturations (lowest peripheral oxygen saturation was 58%) requiring repeated stimulations, oxygen supply, and the removal of the transdermal patch (oral MED: 0.308 mg/kg/d [33], equivalent to 21.56 mg/d in a 70kg patient). In the remainder of the inpatient course, we carefully titrated the opioid dose by cutting the matrix patch into quarters [34]. A dose increase to a 5 µg/h transdermal patch was initiated but reduced to 3.75 µg/h due to repeated fatigue.

## 4. Discussion

The literature review shows that SNI is a risk factor for OIRD in pediatric patients. Five out of six articles revealed significant differences in the reported incidence, odds ratio, or relative risk of OIRD for SNI patients. Polypharmacy and disease-associated comorbidities were identified as additional risk factors. The described case reports indicate that the aforementioned risk factors also apply to adolescents and young adults with SNI who receive low-dose oral or transdermal opioid therapy.

The three described clinical cases received a comedication of between 10 and 13 substances. These included 2 to 5 CNS depressants like anticonvulsants, antipsychotics, and sedatives. Polypharmacy is common both in patients with SNI and in PPC patients in general [4,35]. Patients presenting to the focused PPC unit receive an average of six different drugs during their stay (range 0–16). Many of these drugs are central nervous system agents like anticonvulsants, sedatives, barbiturates, and muscle relaxants [6]. Polypharmacy bears the risk of drug interaction and enhancement of depressant effects on the central nervous system, such as when combining sedating medication like benzodiazepines with mu-opioid agonists [29,30]. In this patient group, regular medication reconciliations by a hospital pharmacist and medication use reviews should be performed [36,37].

We discussed four different opioids in the case reports: morphine, levomethadone, tilidine, and buprenorphine. Methadone is a racemic mixture of the L-isomer, the active analgesic form, and the R-isomer with unknown action. The L-isomer is available in some countries as levomethadone. It is twice as strong as methadone with potentially less severe side effects (especially a prolongation of the QT interval), which is mainly attributed to the R-isomer. Tilidine is available only in a few countries like Belgium, Germany, Spain, and South Africa [38]. There is just an oral preparation of tilidine. A fixed combination with naloxone is oftentimes administered to prevent abusive intravenous use [39]. The strength of tilidine is comparable to that of the weak opioid tramadol. The therapeutic activity of tilidine is mainly related to its active metabolite nortilidine, which penetrates the blood-brain barrier and binds to the µ-opioid receptor as a potent agonist. Nortilidine concentration depends on CYP3A4 activity [40]. In PPC, tilidine is prescribed because of its low potential to worsen preexisting epilepsy.

In contrast to other opioids, buprenorphine has shown a ceiling effect regarding OIRD in healthy adult volunteers [41]. Interestingly, the patient in *Case 3* showed oxygen desaturations under transdermal buprenorphine therapy. The obstructive sleep apnea and the CNS depressant effect of the patient’s anticonvulsive comedication may have contributed to the OIRD.

Another potential cause of OIRD named by Jay et al. [27] is the frequent disease-associated comorbidities of SNI patients. These patients often have an impaired respiratory drive, cardiorespiratory comorbidities, chronic obstruction of the upper airways, neuromuscular and postural abnormalities, or gastroesophageal reflux, which may impact the patient’s respiration and possibly lead to aspiration. The sensation of pain has also been observed in relation to impaired respiration function, as it increases the tonic input to the respiratory centers. In patients with already weakened breathing and who are suffering from severe pain, analgesia can lead to loss of respiratory drive and resulting respiratory depression [24].

Increased sensitivity to opioids in patients with recurrent hypoxemia before treatment can also result in respiratory depression [42]. Treating these patients hence requires careful dose titration. During the opioid titration, pain is best to be monitored with standardized tools like the pediatric pain profile for chronic pain or the revised FLACC scale for postoperative pain [43,44]. In contrast, hyperoxaemia due to supplemental oxygen during opioid treatment resulted in more frequent respiratory depressions in healthy young adults [45]. These findings might have clinical implications for pediatric patients when receiving supplemental oxygen to avert possible decreases in oxygen saturation, although sole dependence on oxygen saturation monitoring for OIRD alone might give a false sense of security. Therefore, multimodal monitoring, including clinical appearance and vital signs, is necessary [20]. Non-invasive capnography potentially detects respiratory depression in children and adults earlier than pulse oximetry [46,47,48]. However, an additional nasal cannula/face mask may not be suitable for the palliative care setting, where normalcy is a recurrent wish voiced by families [49]. Thus, monitoring respiratory rate changes probably provides the greatest predictability of imminent respiratory depression and failure [20].

Two patients of the reported cases were severely underweight. Nutrition is a widespread problem in pediatric patients with SNI [50]. According to Chidambaran et al. [29], opioid-induced critical respiratory events are quite likely in underweight patients. The authors theorized that low serum albumin levels lead to higher free concentrations of opioids, which, in turn, provokes severe side effects. This mechanism could especially apply to highly lipophilic opioids like fentanyl, buprenorphine, or methadone [51].

However, studies on cachectic adult cancer patients with low plasma albumin concentrations have reported a negative effect on transdermal absorption of fentanyl [52]. Even if the clinical implications of these findings remain unclear for pediatric or PPC patients, knowledge of opioid pharmacokinetics may prevent potential side effects and increase therapeutic efficacy.

Symptom control is a primary outcome domain for PPC patients [49]. In the focused specialized PPC unit, starting treatment with strong opioids is the most frequent pharmacological intervention [6]. In order to achieve the best attainable quality of life for a pediatric patient with LTC/LLC, it is ethically acceptable to take on the low risk of respiratory depression in order to relieve pain or severe dyspnea [53]. For clinical practice, this implies that careful assessment should be done regarding not only the symptom to be treated but also the comorbidities or medications that may affect opioid therapy-related side effects. It is important to anticipate potential side effects when opioids are prescribed for distressing symptoms in children with LTC/LLC. Caregivers such as parents as well as non-specialist physicians and nurses need to be educated about common (e.g., constipation, nausea or vomiting) and rare side effects (e.g., OIRD, cerebral fits, itch) of opioids. The detection and management of possible adverse effects must become routine concerning pharmacological treatment with strong opioids in inpatient PPC. These principles can then be the foundation of effective and safe symptom control.

## 5. Conclusions

Children, adolescents, and young adults with SNI have an increased risk of OIRD. As our case reports show, this risk also seems applicable to the oral or transdermal administration of opioids preferred in PPC.

Although the risk of OIRD is relatively low when starting opioid treatment, this vulnerable cohort of patients requires special attention. Inpatient care, if available in a specialized PPC unit, makes it possible to assess the effects and side effects of opioid therapy. These side effects must be anticipated and treated proactively. We recommend a careful titration, starting from a third of the starting dose of otherwise healthy opioid-naïve children.

Dose determination includes close monitoring of the patient’s vigilance, respiratory rate, pulse oximetry, and a prepared emergency plan (e.g., for naloxone). It is highly recommended to become aware of the additional risk linked to disease-associated comorbidities and drug interactions, especially in the case of polypharmacy. The simultaneous treatment of opioids and CNS depressants should be initiated with utmost caution and only in conjunction with opioid dose reduction. Concerns about possible OIRDs should never be a barrier to adequate symptom relief in PPC patients. If feasible, a palliative expert should be consulted.

The majority of patients described in the reviewed articles received short-term intravenous opioids for postoperative pain. However, the results should not simply be generalized to the population of pediatric patients with SNI suffering from chronic pain. Future research should focus on assessing and managing pain in pediatric patients with SNI and their risk of OIRD, especially in PPC. As a limitation of the study, concentration on the (medical) database PubMed/Medline must be mentioned. Even if this is sufficient for the purpose of the scoping review, it cannot be excluded that relevant literature is available in other databases. Therefore, conducting a systematic review is worth considering in the future. Improvement of PPC through research and clinician education can reduce the risks of both symptom undertreatment and OIRD [53,54,55,56].

## Figures and Tables

**Figure 1 children-07-00312-f001:**
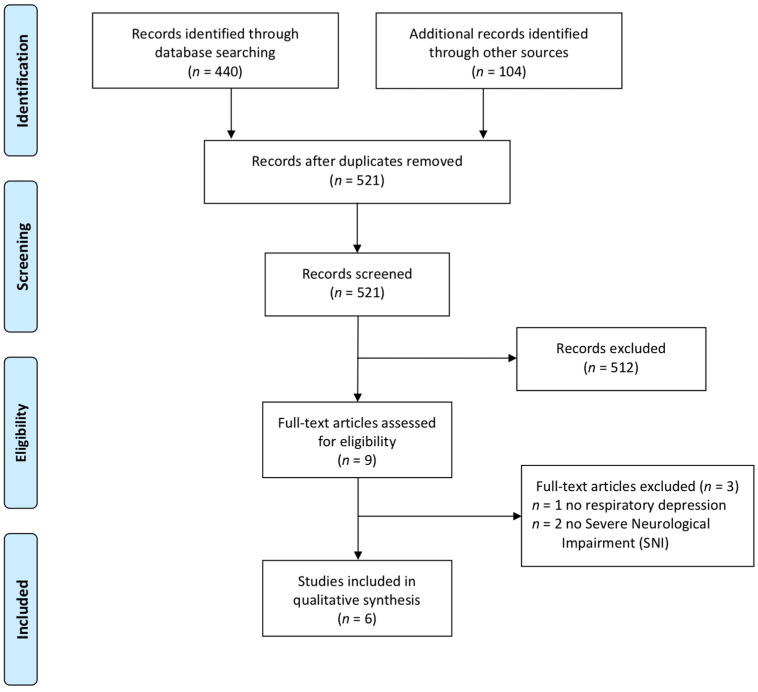
Flow chart of literature eligibility.

**Table 1 children-07-00312-t001:** Search strategy.

opioid [MeSH Terms] OR opioid OR morphine [MeSH Terms] OR morphine OR morphine derivatives [MeSH Terms] OR Fentanyl [MeSH Terms] OR Fentanyl OR Buprenorphine [MeSH Terms] OR Buprenorphine OR Methadone [MeSH Terms] OR Methadone OR Levomethadone OR Tramadol [MeSH Terms] OR Tramadol OR Tilidine [MeSH Terms] OR Tilidine	AND	Apnea [MeSH Terms] OR apnea OR “respiratory depression”	AND	neuromuscular diseases [MeSH Terms] OR neuromuscular diseases OR neurodegenerative diseases [MeSH Terms] OR neurodegenerative diseases OR neurotoxicity Syndromes [MeSH Terms] OR neurotoxicity Syndromes OR neurological manifestations [MeSH Terms] OR neurological manifestations OR “psychomotor impairment” OR “severe psychomotor impairment” OR “developmental disabilities”	AND	child [MeSH Terms] OR child * OR Adolescent [MeSH Terms] OR Adolescent * OR infant [MeSH Terms] OR Infant * OR pediatrics [MeSH Terms] OR Pediatric *

**Table 2 children-07-00312-t002:** Studies reporting on Opioid-Induced Respiratory Depression (OIRD) in children and adolescents with Severe Neurological Impairment (SNI).

Reference	Study Design	Objective	Sample Description, Size, Age	Definition of Critical Respiratory Events	Risk Factors and Main Results
Voepel-Lewis (2008) [26]	Retrospective chart review	Comparing the prevalence of clinically significant adverse events in children receiving Patient-Controlled Analgesia (PCA) vs. Patient-Controlled Analgesia by Proxy (PCAP) after surgery.	Random sample of children receiving PCA and PCAP. *n* = 157 PCA age (years) 13.1 ± 3.2 *n* = 145 PCAP age (years) 7.08 ± 5.3	Respiratory depression requiring rescue events: Treated with naloxone, airway management, or escalation of care	Opioid dose and cognitive impairment were independent predictors of rescue events irrespective of patients receiving PCA or PCAP. Despite reduced opioid consumption, the odds ratio for a rescue event in patients with cognitive impairment was 2.4 (CI 1.3–4.2). Additional risk factors associated included orthopedic surgery, respiratory comorbidity, continuous basal opioid infusion, diazepam use, and higher opioid doses.
Jay (2017) [27]	Retrospective cohort study	Quantification of the risks and effectiveness of Nurse-Controlled Analgesia (NCA) for postoperative pain in children with neurodevelopmental disabilities compared to a control group.	Patients who received NCA and were identified from the clinical patient record as having neurodevelopmental disabilities were divided into a neuro-developmental disabilities group (NDG) and a control group (CG). *n* = 12904 age distribution not reported	Respiratory depression defined as a depression of the respiratory rate below an age-defined rate	The cumulative incidence of OIRD in the neurodevelopmental disability group was 1.09% vs. 0.59% in the control group [odds ratio 1.8 (98% chance that the true odds ratio was >1)]. Significant interactions between postoperative morphine dose and SNI were shown in a logistic regression model. Children with cerebral palsy, Down’s syndrome, and encephalopathy were at the highest risk of developing respiratory depression in the neurodevelopmental disability group. Children with SNI were suspected of having an increased risk of respiratory depression because of an increased incidence of impaired respiratory drive, cardiorespiratory deficits, neuromuscular and postural abnormalities, and gastroesophageal reflux.
Czarnecki (2008) [28]	Retrospective chart review	Evaluation of outcomes associated with mainly postoperative Parent/Nurse- controlled Analgesia (PNCA) in pediatric patients with identified developmental delay.	Patients who received PNCA and were identified as being developmentally delayed based on clinical documentation. *n* = 71 age (years) 9.9 ± 5.28	Requirement of naloxone for sedation or respiratory depression.	2.8% of the patients received naloxone to treat side effects of opioids (oversedation or respiratory depression). Adjuvant sedating medications (diazepam, droperidol, chloral hydrate, and diphenhydramine) may have contributed to respiratory depression.
Chidambaran (2014) [29]	Retrospective chart review	Naloxone usage for opioid-induced critical respiratory events in children as a quality measure of opioid safety in patients receiving postoperative and other opioid therapy.	All patients who received naloxone for opioid-induced critical respiratory events. *n* = 38 age (years) 8.7 ± 8.0	Requirement of naloxone for respiratory depression.	Age <1 year, underweight, obesity, history of prematurity syndrome, developmental delay, obstructive sleep apnea, and respiratory, hepatic, and neurological comorbidities were significant risk factors for early respiratory depression associated with opioid treatment. The unadjusted odds ratios for the need for administration of naloxone were 3.24 (CI 1.36–7.47) for the presence of a syndrome, 4.99 (CI 2.17–11.15) for developmental delay, and 3.87 (CI 1.83–8.07) for neurological impairment.
Morton (2010) [30]	Prospective cohort study	Determination of the incidence, nature, and severity of serious clinical incidents associated with continuous opioid infusion, Patient-Controlled Analgesia (PCA), and Nurse-Controlled Analgesia (NCA) in pediatric patients.	Sample of all pediatric patients who received opioid-infusion, PCA, and NCA. *n* = 10726 age < 1 month = 344 age 1 month–1 year = 1383 age 1–8 years= 3433 age 8–18 years = 5566	Death or permanent harm/ harm but full recovery leading to discontinuation of the technique or requiring significant intervention/ potential but no actual harm.	Eight of the fourteen reports of respiratory depression received naloxone; they were all very young or had significant neurodevelopmental, respiratory, or cardiac comorbidities. Avoidance of concurrent sedatives or opioids and awareness of comorbidities can improve patient safety.
Monitto et al. (2000) [31]	Prospective cohort study	Determination of patient demographics, analgesia effectiveness, and the incidence of complications in pediatric patients receiving Parent/Nurse- Controlled Analgesia (PNCA).	All patients <6 years of age who received PNCA. *n* = 212 age (years) 2.3 ± 1.7	Apnea or oxygen desaturation	No specific risk factor was associated with naloxone administration in nonsurgical and postoperative children under six years of age. Patients’ clinical characteristics were found to be predisposing factors for excessive sedation or respiratory compromise. In this context, additional sedatives, development delay, and congenital anomalies were named.

OIRD: Opioid-Induced Respiratory Depression. Data reported as *n* or mean ± SD.

**Table 3 children-07-00312-t003:** Patient characteristics.

**Case 1**	
Underlying disease and main symptoms	Hypoxic-ischemic encephalopathy Preterm birth at 26 weeks Cardiopulmonary resuscitation in severe sepsis at age 16 Spastic cerebral palsy (GMFCS level V) Contractures Thoracic scoliosis Dislocated hip dysplasia Symptomatic epilepsy with epileptic spasms Hypothyroidism Underweight
Age [years]	18.8
Weight [kg]	33.2
Length [cm]	140
BMI [kg/m²] (Percentile)	16.8 (2.)
Indication for opioid therapy	Musculoskeletal pain
Opioid medication (Route of administration)	Morphine (Enteral)
Dosage	4 × 2 mg
Daily oral morphine equivalent dose [mg/kg/d]	0.241
Relevant comedication during OIRD	Levetiracetam, valproate, dronabinol, baclofen, chloral hydrate, melatonin, ibuprofen
Opioid-induced respiratory event	Hypopnea (lowest respiratory rate 7/min) Oxygen desaturation (lowest oxygen saturation 82%)
OIRD at day of opioid treatment	2
Interventions	Repeated stimulation Oxygen supply Termination of opioid treatment Opioid switch to levomethadone
**Case 2**	
Underlying disease and main symptoms	Early infantile epileptic encephalopathy (Ohtahara syndrome) Obstructive sleep apnea Spastic cerebral palsy (GMFCS level V) Contractures Respiratory failure type I
Age [years]	15
Weight [kg]	23.5
Length [cm]	105
BMI [kg/m²] (Percentile)	21.3 (68)
Indication for opioid therapy	Severe neuroirritability with pain-like behavior
Opioid medication (Route of administration)	Levomethadone (Enteral)
Dosage	3 × 1 mg
Daily oral morphine equivalent dose [mg/kg/d]	1.02 [33]
Relevant comedication during OIRD	Lamotrigine, risperidone, domperidone, melatonin
Opioid-induced respiratory event	Apnea Oxygen desaturation (lowest oxygen saturation 88%)
OIRD at day of opioid treatment	8
Interventions	Repeated stimulation Oxygen supply Initial termination of opioid treatment
**Case 3**	
Underlying disease and main symptoms	Lissencephaly Ventriculoatrial shunt Focal seizures Spastic cerebral palsy (GMFCS level V) Contractures Thoracic scoliosis Hip dysplasia Hypothyroidism
Age [years]	20.6
Weight [kg]	38.9
Length [cm]	170
BMI [kg/m²] (Percentile)	13.8 (<1)
Indication for opioid therapy	Musculoskeletal pain
Opioid medication (Route of administration)	Buprenorphine (Transdermal)
Dosage	5 µg/h
Daily oral morphine equivalent dose [mg/kg/d]	0.308 [33]
Relevant comedication during OIRD	Oxcarbazepine, valproate, metamizole
Opioid-induced respiratory event	Repeated oxygen desaturation (lowest oxygen saturation 58%)
OIRD at day of opioid treatment	1
Interventions	Repeated stimulation Oxygen supply Initial termination of opioid treatment Dose reduction

GMFCS: Gross Motor Function Classification System; OIRD: Opioid-Induced Respiratory Depression.

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
