# Peer review of "Opioid-Induced Respiratory Depression in Pediatric Palliative Care Patients with Severe Neurological Impairment—A Scoping Literature Review and Case Reports"

_children, 2020, doi:10.3390/children7120312_

Round 1

Reviewer 1 Report

Thank you for the opportunity to review the article, “Opioid-Induced Respiratory Depression (OIRD) in Pediatric Palliative Care Patients with Severe Neurological Impairment (SNI) – A Scoping Literature Review and Case Reports.” This well-written manuscript presents a summary of the limited current literature on OIRD in the population of children with SNI, as well as 3 case reports of ORID in the authors’ home institution. The topic is important, especially because—as the authors point out—the bulk of research investigating pediatric pain management typically excludes children with SNI. This population, however, is among the most vulnerable to pain management strategies and downstream consequences, like the adverse event of OIRD described in this manuscript.

A few specific comments and suggestions:

Lines 43-44: How was the SNI definition operationalized for purposes of the literature review? The authors may want to add an additional column in Table 2 to describe the study population for each referenced study, since some of the studies are focused completely on those with medical complexity (reference #28), while others report on a broad population of children (reference #30).

Line 56: Does a common research definition of OIRD exist, perhaps based on vital signs, rescue medications, diagnoses? If so, it may be worthwhile to define here for the reader, since it is relevant to both the literature review and the 3 presented cases. While Table 2 does include a column for “Definition of Critical Respiratory Events,” I still think an early definition is useful to provide to reassure readers that included/excluded records in Figure 1 were all evaluated against a common threshold for the outcome.

Lines 71-73: These sentences outline the research gap addressed in this scoping review. I wonder if this could be moved earlier in the introduction (and perhaps added to the abstract) to quickly establish the need for this review.

Line 93: Why were only certain specific medications included in the search and not others, e.g. oxycodone?

Lines 94-96: Were children who received respiratory support (trach/vent, intubation, etc.) included or excluded?

Line 114: The text presented in the final column in Table 2 did not always line up exactly with the relevant study and this made it difficult to understand which study was connected to which risk factor/result findings. Perhaps the authors could add horizontal row lines between the 6 studies?

Lines 174-180 and Lines 246-253: Given the polypharmacy present for many of the described patients, is there a role for medication therapy management and/or involvement of pharmacist review?

Lines 290-294: Could the authors make a brief recommendation about the best way(s) to monitor pain and response to analgesia in children with SNI who may have impaired communication?

Lines 313-315: Thank you for making this important distinction between acute and chronic pain.

Author Response

Point 1: Lines 43-44: How was the SNI definition operationalized for purposes of the literature review? The authors may want to add an additional column in Table 2 to describe the study population for each referenced study, since some of the studies are focused completely on those with medical complexity (reference #28), while others report on a broad population of children (reference #30).

Response 1: Thank you very much for your remarks. We have clarified the definition of SNI in the introduction (lines 44-46) and in the section "methods and materials" (lines 93-94) using the delphi consensus-based definition by Allen et al. (Ref. 7. Allen, J.; Brenner, M.; Hauer, J.; Molloy, E.; McDonald, D. Severe Neurological Impairment: A delphi consensus-based definition. Eur J Paediatr Neuro 2020, doi:10.1016/j.ejpn.2020.09.001).

Furthermore, we have now described the patient populations under "study" in table 2 under "Sample description, size, age".

Point 2: Line 56: Does a common research definition of OIRD exist, perhaps based on vital signs, rescue medications, diagnoses? If so, it may be worthwhile to define here for the reader, since it is relevant to both the literature review and the 3 presented cases. While Table 2 does include a column for "Definition of Critical Respiratory Events," I still think an early definition is useful to provide to reassure readers that included/excluded records in Figure 1 were all evaluated against a common threshold for the outcome.

Response 2: Thank you very much for this important note. Unfortunately, there is no common definition for OIRD. For the purpose of this scoping review, OIRD was defined as any opioid-related significant decrease in the respiratory rate or oxygen desaturation, which led to apnea or the need for medical intervention (lines 91-92). Because of the lack of a universal definition, each study's definitions have been included in Table 2.

Point 3: Lines 71-73: These sentences outline the research gap addressed in this scoping review. I wonder if this could be moved earlier in the introduction (and perhaps added to the abstract) to quickly establish the need for this review.

Response 3: Thank you very much for your feedback. The research gap was one of the intentions to write this scoping review. We have moved the paragraph to an earlier part of the introduction and added the fact to the abstract.

Point 4: Line 93: Why were only certain specific medications included in the search and not others, e.g. oxycodone?

Response 4: The search terms included "opioids" in general and the specific opioids regularly used in PPC in Germany. Oxycodone has severe disadvantages (the risk of prolongation of the QT interval and the development of dangerous ventricular tachycardia) and no advantage in comparison to other opioids (Behzadi M, Joukar S, Beik A. Opioids and Cardiac Arrhythmia: A Literature Review. Med Princ Pract. 2018;27(5):401-414) therefore it is seldom used in Germany. But, regarding our literature search, the MeSH term "opioid", as well as the extensive hand search, included all opioids. We added one sentence in lines 112-113.

Point 5: Lines 94-96: Were children who received respiratory support (trach/vent, intubation, etc.) included or excluded?

Response 5: We have not explicitly excluded children who have a tracheostomy or respiratory support. We have added this to our methods and materials section in lines 118-119.

Point 6: Line 114: The text presented in the final column in Table 2 did not always line up exactly with the relevant study and this made it difficult to understand which study was connected to which risk factor/result findings. Perhaps the authors could add horizontal row lines between the 6 studies?

Response 6: We have added horizontal lines to the table to improve readability of Table 2.

Point 7: Lines 174-180 and Lines 246-253: Given the polypharmacy present for many of the described patients, is there a role for medication therapy management and/or involvement of pharmacist review?

Response 7: Thank you for this splendid comment. We added a sentence in lines 287- 288: " In this patient group, regular medication reconciliations by a hospital pharmacist and medication use reviews should be performed".

Point 8: Lines 290-294: Could the authors make a brief recommendation about the best way(s) to monitor pain and response to analgesia in children with SNI who may have impaired communication?

Response 8: Thank you very much for this suggestion. We added a sentence in lines 303- 304 "During the opioid titration, pain is best to be monitored with standardized tools like the pediatric pain profile for chronic pain or the revised FLACC scale for postoperative pain".

Point 9: Lines 313-315: Thank you for making this important distinction between acute and chronic pain.

Response 9: Thank you.

Reviewer 2 Report

Nice article. No edits recommended

Author Response

Point: Nice article. No edits recommended

Response: Thank you very much. We are pleased that you enjoyed the article.

Reviewer 3 Report

Overall a well written paper on an important but little written about topic.  The authors bring balance to the topic (identifying risk, but also mandating the needing to treat symptoms in children with severe neurological impairment).

The authors demonstrated their search methodology and use the PRISMA-ScR schema.  How many of the reviewers reviewed each abstract - was it just once, and was their assessment blinded to each other?

Are there other databases other then Medline that could be searched?  If so, could the authors mention this as a limitation.

Could the authors please consider answer the following question for me.  Is there scope for more rapid dose titration of opioids for an inpatient - where a close multi-modal method of continuous observation can be provided, compared to at home, where dose titration might be more judicious and cautious?

Author Response

Point 1: Overall a well written paper on an important but little written about topic. The authors bring balance to the topic (identifying risk, but also mandating the needing to treat symptoms in children with severe neurological impairment).

Response 1: Thank you very much. We are pleased that you enjoyed the article.

Point 2: The authors demonstrated their search methodology and use the PRISMA-ScR schema. How many of the reviewers reviewed each abstract - was it just once, and was their assessment blinded to each other?

Response 2: The abstracts were blindly screened by two authors. Discrepancies in the decision of inclusion and exclusion criteria were discussed between the two reviewers. We have specified this in more detail in lines 120-122.

Point 3: Are there other databases other then Medline that could be searched? If so, could the authors mention this as a limitation.

Response 3: Thank you very much for this important note. We have added the exclusive search in PubMed/Medline to the discussion as a limitation of the study (lines 354-357: "As a limitation of the study, concentration on the (medical) database PubMed/Medline must be mentioned. Even if this is sufficient for the purpose of the scoping review, it cannot be excluded that relevant literature is available in other databases. Therefore, conducting a systematic review is worth considering in the future".

Point 4: Could the authors please consider answer the following question for me. Is there scope for more rapid dose titration of opioids for an inpatient - where a close multi-modal method of continuous observation can be provided, compared to at home, where dose titration might be more judicious and cautious?

Response 4: Due to the described potential of side effects up to OIRD, we generally recommend an inpatient stay to start treatment with strong opioids for this particular patient population. In the case reports, it has been shown that an OIRD can be prevented by increasing the dose particularly slowly in some instances.